# Biosynthesized ZnO-NPs from *Morus indica* Attenuates Methylglyoxal-Induced Protein Glycation and RBC Damage: In-Vitro, In-Vivo and Molecular Docking Study

**DOI:** 10.3390/biom9120882

**Published:** 2019-12-16

**Authors:** Satish Anandan, Murali Mahadevamurthy, Mohammad Azam Ansari, Mohammad A. Alzohairy, Mohammad N. Alomary, Syeda Farha Siraj, Sarjan Halugudde Nagaraja, Mahendra Chikkamadaiah, Lakshmeesha Thimappa Ramachandrappa, Hemanth Kumar Naguvanahalli Krishnappa, Ana E. Ledesma, Amruthesh Kestur Nagaraj, Asna Urooj

**Affiliations:** 1Department of Studies in Food Science and Nutrition, University of Mysore, Manasagangotri, Mysuru, Karnataka 570006, India; satishanandan84@gmail.com (S.A.); farhasiraj7@gmail.com (S.F.S.); 2Applied Plant Pathology Laboratory, Department of Studies in Botany, University of Mysore, Manasagangotri, Mysuru, Karnataka 570006, India; botany.murali@gmail.com (M.M.); hemanthbot@gmail.com (H.K.N.K.); dr.knamruthesh@botany.uni-mysore.ac.in (A.K.N.); 3Department of Epidemic Disease Research, Institute for Research & Medical Consultations (IRMC), Imam Abdulrahman Bin Faisal University, P.O. Box 1982, Dammam 31441, Saudi Arabia; 4Department of Medical Laboratories, College of Applied Medical Sciences, Qassim University, Qassim 51431, Saudi Arabia; dr.alzohairy@gmail.com; 5National Center for Biotechnology, Life Science and Environmental Research Institute, King Abdulaziz City for Science and Technology, Riyadh P.O. Box 6086, Saudi Arabia; malomary@kacst.edu.sa; 6Department of Studies in Zoology, University of Mysore, Manasagangotri, Mysuru, Karnataka 570006, India; hnsarjan@gmail.com; 7Department of Studies in Biotechnology, University of Mysore, Manasagangotri, Mysuru, Karnataka 570006, India; lakshmeeshat6@gmail.com; 8Centro De Investigación en Biofísica Aplicada y Alimentos, Universidad Nacional de Santiago del Estero (CIBAAL-UNSE-CONICET), FCEyT, RN 9, km 1125, CP 4206 Santiago del Estero, Argentina; anael@unse.edu.ar

**Keywords:** glycation, in-vivo, diabetes, methylglyoxal, molecular docking, ribose, ZnO-NPs

## Abstract

The development of advanced glycation end-products (AGEs) inhibitors is considered to have therapeutic potential in diabetic complications inhibiting the loss of the biomolecular function. In the present study, zinc oxide nanoparticles (ZnO-NPs) were synthesized from aqueous leaf extract of *Morus indica* and were characterized by various techniques such as ultraviolet (UV)-Vis spectroscopy, Powder X-Ray Diffraction (PXRD), Fourier Transform Infrared Spectroscopy (FT-IR), Scanning electron microscopy (SEM), and energy dispersive spectroscopy (EDS). Further, the inhibition of AGEs formation after exposure to ZnO-NPs was investigated by in-vitro, in-vivo, and molecular docking studies. Biochemical and histopathological changes after exposure to ZnO-NPs were also studied in streptozotocin-induced diabetic rats. ZnO-NPs showed an absorption peak at 359 nm with a purity of 92.62% and ~6–12 nm in size, which is characteristic of nanoparticles. The images of SEM showed agglomeration of smaller ZnO-NPs and EDS authenticating that the synthesized nanoparticles were without impurities. The biosynthesized ZnO-NPs showed significant inhibition in the formation of AGEs. The particles were effective against methylglyoxal (MGO) mediated glycation of bovine serum albumin (BSA) by inhibiting the formation of AGEs, which was dose-dependent. Further, the presence of MGO resulted in complete damage of biconcave red blood corpuscles (RBCs) to an irregular shape, whereas the morphological changes were prevented when they were treated with ZnO-NPs leading to the prevention of complications caused due to glycation. The administration of ZnO-NPs (100 mg Kg^−1^) in streptozotocin(STZ)-induced diabetic rats reversed hyperglycemia and significantly improved hepatic enzymes level and renal functionality, also the histopathological studies revealed restoration of kidney and liver damage nearer to normal conditions. Molecular docking of BSA with ZnO-NPs confirms that masking of lysine and arginine residues is one of the possible mechanisms responsible for the potent antiglycation activity of ZnO-NPs. The findings strongly suggest scope for exploring the therapeutic potential of diabetes-related complications.

## 1. Introduction

Advanced glycation end products (AGEs) are heterogeneous compounds that are formed when protein or lipid exposed to sugar in the bloodstream. The factors responsible for AGEs formation include aging, degenerative diseases such as diabetes, atherosclerosis, chronic kidney disease, Alzheimer’s disease, etc. AGEs are produced by a sequence of events (Maillard reaction), wherein the sugars react with free amino groups of peptides, proteins, and amino acids specifically with lysine and arginine residues to form a ketoamine known as Amadori product, which is associated with the diabetic complications[1,2,3,4,5].

Advanced glycation endproducts lead to chemical modifications in the targeted protein which, in turn, undergoes specific changes such as detachment of ligands and oxidation of thiol group leading to the formation of disulfide linkage or thiol radicals [6]. The glycated protein form adducts with other native proteins, which induce protein aggregation collectively leading to oxidative stress resulting in the damage of red blood corpuscles (RBCs) [7]. Presently, aminoguanidine (AG) is used as a standard drug in the inhibition of AGEs which are associated with diabetic complications both under in-vitro and in-vivo conditions [8]. AGEs inhibitors such as pyridoxamine [9], tenilsetam [10], and metformin [11] have been tried in the treatment of diabetes without much success, due to their relatively low efficacies and poor pharmacokinetics [12], however, AG has been found effective in AGEs inhibition but also proved to be highly toxic for diabetic patients [13].

Hence, researchers are focusing on the active ingredients, which are not affecting the native structure of biomolecules and may be used as potential inhibitors of AGEs. Nowadays, nanotechnology has begun as an interdisciplinary research area involving physics, chemistry, biology, and medicine with immense potential for initial accurate detection of diseases and treatment [14,15]. Due to its smaller size (1 to 100 nm) when compared to cells, these particles can interact with both intra and inter cellularly biomolecules [16]. Biosynthesised ZnO-NPs have been reported to possess better antimicrobial activity when compared with chemically synthesized nanoparticles. Among the metal oxide nanoparticles, ZnO-NPs have been used extensively in biological applications due to their non-toxic nature, and they are also listed as “generally recognized as safe” (GRAS) by the U.S. FDA (21CFR182.8991).

Only a few studies have been reported to date on nanoparticles against inhibition of glycation. It has been reported that the size and concentration of the gold nanoparticles allowed variations in the total surface area of the colloidal suspensions influencing the extent modification of bovine serum albumin (BSA) with d-ribose [17], while selenium nanoparticles protected structural modifications of proteins during glycation [18]. Similarly, cerium dioxide nanoparticles were able to protect the lens epithelial cells from the damage of oxidative stress by scavenging reactive oxygen species (ROS) and further attenuate α-crystallin glycation by possessing antioxidant and antiglycation properties [19]. In addition, the previous study on ZnO-NPs synthesized from aqueous extract of *Aloe vera* leaf was found to be a potent antiglycation agent, as they could inhibit the formation of AGEs and protect the protein structure from modification [16]. The reports suggest that nanoparticles affect the protein structure differently which might be influenced by various factors, including size and concentration.

*Morus indica* L. is used as traditional medicine for its hypoglycemic and diuretic properties. In our laboratory, varieties of *M. indica* have been screened for its proximate composition, phytochemical profile, antioxidant, anti-hypercholesterolemic, anti-cancer and anti-diabetic effect in in-vitro and ex-vivo models [20]. The earlier studies using the extract of *M. indica* has shown a potential anti-glycation effect in the BSA-glucose model [21]. Hence, the present study was aimed at the biosynthesis of ZnO-NPs from *M. indica* and to evaluate its inhibitory efficacy against AGEs formation. Based on our previous reports, the current work was aimed at the biosynthesis of ZnO-NPs from *M. indica* and to assess its inhibitory efficacy against AGEs formation.

## 2. Materials and Methods

### 2.1. Collection of Plant

The *Morus indica* leaves of G4 (ISGR Reg. No. 050564) were collected from CSRTI (Central Sericulture Research and Training Institute), Mysuru, in the month of May 2016 and used to biosynthesize ZnO-NPs.

### 2.2. Chemicals

Bovine serum albumin (purity > 98%), methylglyoxal (40% in H_2_O), acetylglycyl-lysine methyl ester (G.K.) peptide (purity > 98%), aminoguanidine hydrochloride (purity > 98%), zinc oxide nanopowder were obtained from SRL (India). Nile red and δ-Gluconolactone were purchased from HiMedia (India) and all other chemicals used in the study were of analytical grade.

### 2.3. Biosynthesis of ZnO-NPs

ZnO-NPs were synthesized by the solution combustion method according to Murali et al. [22] with minor modifications. Fresh leaves (30 g) of *M. indica* were collected and washed with running tap water and subsequently blended with a hand blender using 300 mL of sterile distilled water and filtered through Whatman No. 1 filter paper. About 20 mL of the plant extract was heated on a magnetic stirrer and when the temperature reached about 60–80 °C, 2 g of zinc nitrate hexahydrate was added little by little with constant stirring with magnetic beads until the solution turned to paste. The obtained paste material was placed in a furnace maintained at 400 °C for 2hand the obtained powder was subjected for physico-chemical characterization.

### 2.4. Characterization of Biosynthesized ZnO-NPs

Ultraviolet (UV)-Vis spectra (Beckman Coulter, DU739, Krefeld, Germany) and Powder X-Ray Diffraction (PXRD) patterns of ZnO-NPs were analyzed as reported by Ashraf et al. [23] and the particle size was calculated using the Scherrer’s formula:
$$\Phi =\frac{K\lambda }{\beta \cos\theta },$$
where Φ is the crystalline size, λ is the wavelength of *X*-Ray used. K is the shape factor, β is the full line width at the half maximum (FWHM) elevation of the main intensity peak, and θ is the Bragg angle. Fourier Transform Infrared (FT-IR) spectroscopy (Perkin Elmer, Waltham, MA 02451 USA) was carried out to know the functional groups. Scanning electron microscopy (SEM) (HITACHI, S-3400N, Tokyo, Japan) and energy dispersive spectroscopy (EDS) of biosynthesized ZnO-NPs were performed as a method described by Jalal et al. [24].

### 2.5. Effect of Biosynthesized ZnO-NPs from M. indica on Protein Glycation under In-Vitro

In the protein glycation studies, aminoguanidine (10 mM) was used as a positive control. The inhibition of protein glycation activity was compared between commercially available Zinc oxide nano-powder and biosynthesized ZnO-NPs from *M. indica*.

### 2.6. Effect of ZnO-NPs on the Formation of Amadori Product

#### Haemoglobin-*δ*-Gluconolactone (*δ*-Glu) Assay

The biosynthesized ZnO-NPs were evaluated for their inhibitory effect on the formation of Amadori products following the method of Losso et al. [25]. About 0.2 mL of fresh human blood containing ZnO-NPs (1, 2.5 and 5 mg mL^−1^ dissolved in phosphate buffer saline of pH 7.4) along with *δ*-gluconolactone (*δ*-glu) (50 mM dissolved in phosphate buffer saline (PBS, pH 7.4). At 37 °C for 16 h the reaction mixture (1 mL) was incubated. The blood sample containing *δ*-glu served as a negative control, while the sample with AG served as a positive control (PC). After incubation, the samples were measured for the percentage of inhibition of HbA1c by using the hemoglobin A1C chromatographic-spectrophotometric ion-exchange kit (Biosystems, Tamil Nadu, India).

### 2.7. Inhibitory Effect of ZnO-NPs on MGO Mediated Protein Glycation

#### 2.7.1. Sample Preparation

Bovine serum albumin (BSA) with methylglyoxal (MGO) was glycated as described by the method of Prasanna and Saraswathi [7] with slight modifications. The reaction mixtures contained BSA (10 mg mL^−1^) with methylglyoxal (10 mM) in a final volume of 1 mL of 0.1 M PBS at pH 7.4. At 37 °C for 24 h the reaction mixture was incubated with different concentrations of ZnO-NPs (1, 2.5, and 5 mg mL^−1^). BSA-MGO with or without AG (10 mM) served as a positive and negative control, while only BSA served as native. The modified BSA was subjected to dialysis (PBS for 24 h) to remove unbound MGO and stored at −20 °C for further use.

#### 2.7.2. Effect of ZnO-NPs on Inhibition of AGEs Formation

The dialyzed sample, as mentioned above, was subjected to evaluate the total formed fluorescence AGEs by excitation (330 nm) and emission (440 nm). Moreover, the formation of argpyrimidine was assessed by AGEs-specific fluorescence by excitation (320 nm) and emission (380 nm) by following the method of Prasanna and Saraswathi [7].

#### 2.7.3. Protective Effect of ZnO-NPs on Red Blood Corpuscles

From a healthy individual, 2 mL of the blood sample was collected in citrate-containing tubes and the preparation of the sample was carried out according to Prasanna and Saraswathi [7]. The blood samples were centrifuged at 1000 rpm for 20 min to separate the RBCs, and the other plasma proteins were decanted. With PBS the pellet containing RBCs was washed thrice and were incubated with MGO (5 μM) in the presence and absence of ZnO-NPs (5 mg mL^−1^), and the reaction was continued for 1 h at 37 °C. The reaction mixture was centrifuged and washed with PBS thrice after incubation to pellet the RBCs. The fixation of obtained RBCs was done with 2.5% glutaraldehyde for 30 min, followed by centrifugation at 1000 rpm for 30 min. To remove the adherence of glutaraldehyde the pellet obtained was washed repeatedly with PBS and was dried overnight by placing it on an aluminum foil. Before using SEM (HITACHI S-3400N, Tokyo, Japan), the samples were coated with gold particles and then analyzed.

### 2.8. Inhibitory Effect of ZnO-NPs on N-Acetylglycyl-Lysine Methyl Ester (G.K.) Peptide Mediated Ribose Glycation

The reaction mixture of 0.5 M sodium phosphate buffer (pH 7.4) containing G.K. peptide (80 mg mL^−1^) with ribose (0.1 M) was used to perform the assay under sterile conditions as described by Losso et al. [25]. About 1 mL reaction mixture containing ZnO-NPs (5 mg mL^−1^) was incubated at 37 °C for 24 h. Samples were analyzed for specific fluorescence of excitation (340 nm) and emission (420 nm) at the end of the incubation. The reaction mixture with or without AG was used as a positive and negative control, respectively.

### 2.9. Effect of Biosynthesized ZnO-NPs from M. indica on Protein Glycation under In-Vivo

#### 2.9.1. Experimental Animals

Healthy adult male Wistar strain rats (200–250 g) were taken for the study and were housed under standard environmental conditions with 12 h (light) and 12 h (dark) cycle in polypropylene cages maintained on commercial rat chow ad libitum. The study was carried out with due approval from the Institutional Animal Ethics Committee of the University of Mysore, Manasagangotri, Mysuru (Animal ethics approval No: UOM/IAEC/05/2017).

#### 2.9.2. Incidence of Induction Streptozotocin of Diabetes and Treatment

Hyperglycemia was induced using streptozotocin (STZ) at 40 mg kg^−1^ body weight by intraperitoneal injection dissolved in 0.1 M citrate buffer (pH 4.5) [26]. The fasting blood glucose level was measured by using a glucometer. Values with >250 mg dL^−1^ were considered to be hyperglycemic rats and used for the experiment. Based on their weights using a randomized block design, the rats were divided into four groups (*n* = 6 in each group) *viz*. Group 1-Healthy control rats, Group 2-STZ-induced hyperglycemic rats without treatment, Group 3-STZ-induced hyperglycemic rats treated with AG (30 mg kg^−1^ body weight by intraperitoneal), Group 4-STZ-induced hyperglycemic rats treated with ZnO-NPs (100 mg kg^−1^ body weight was given by intraperitoneal based on the acute toxicity study) and treated accordingly up to six weeks. The rats were fasted overnight at the end of the study period and anesthetized with diethyl ether.

#### 2.9.3. Biochemical Studies

The serum was separated from the blood samples by centrifugation (3000 rpm at 4 °C for 15 min) and stored at −30 °C, which was collected from the rat’s heart. The blood serum was used to estimate creatinine (CR), bilirubin (BIL), blood urea nitrogen (BUN), total protein (TP), albumin (ALB), alanine aminotransferase (ALT), aspartate aminotransferase (AST), alkaline phosphatase (ALP), serum’s total cholesterol (TC), triglyceride (TG), and high-density lipoprotein (HDL). The test was performed according to the manufacturer’s guidelines by using commercial kits (Agappe Diagnostics Ltd., Kerala, India).

#### 2.9.4. Histopathological Studies

The liver and kidney from anesthetized rats were excised immediately and washed with PBS and weighed. Small portions of the liver and kidney were fixed in 10% formaldehyde and dehydrated with graded series of ethanol (50–100%) followed by infiltration using paraffin. The sections (4–5 µm) obtained through microtome were stained with hematoxylin and eosin dye and examined under a microscope for histopathological changes [27].

### 2.10. Molecular Docking of BSA with ZnO-NPs and AG

The characterization of the binding site between the BSA protein with AG and ZnO nanoparticle was estimated by molecular docking. For ZnO-NPs docking, the structure reported by our previous studies was used [28]. Flat ZnO-NPs surfaces (15 × 25 Å) mimicking the particle curvature of a ZnO-NPs were constructed. ZnO structures were generated using the GaussView software and Gaussian09 program package. The surfaces were optimized using MGL Tools 1.5.6 (The Scripps Research Institute, USA) by adding Marsilli–Gasteiger partial charges on each constituent atom. The surface diameters were estimated using VEGA ZZ Support Pack (Universita degli Studi di Milano, Italy). For AG molecule, their structure was previously optimized using Gaussian09 program package [29]. Docking calculation was performed using AutoDock 4.2 tool [30] with a semi-empirical free-energy force. The crystal structure of BSA was obtained from Protein Data Bank (PDB ID: 4F5S). The PEG molecule together with water molecules present in the structure was removed and then polar hydrogen atoms and Kollman united atom charges were added. An initial grid volume (126 × 126 × 126 with 1 Å grid spacing) covering all surfaces of the protein was used to improve the free movement of all structures around the protein for the evaluation of binding regions. Then a smaller grid volume (60 × 60 × 60) was used to estimate the residues binding, and the AG and ZnO-NPs and BSA structures were treated as rigid docking. Default AutoDock parameters with the Lamarckian genetic algorithm were used. Finally, the best scoring (i.e., with the lowest docking energy or more populated) docked model was chosen to represent the most probable binding mode predicted.

### 2.11. Statistical Analysis

Statistical analysis was performed using SPSS Inc., Chicago, IL, USA 16.0. (Statistical significance for all tests was set at (*p* < 0.05). Further, Tukey’s Honest Significant Differences (HSD) test was used to find means that were significantly different from each other.

## 3. Results and Discussion

### 3.1. Characterization of Biosynthesized ZnO-NPs

From our previous studies, it has been noted that apigenin was detected in the aqueous leaf extract which possesses antiglycation properties. To date, the exact mechanism involved in the formation of ZnO-NPs from plant extracts has not been reported, but it has been quoted that polar groups are responsible for it [31,32,33]. Hence, one of the plausible mechanisms for the capping effect of the plant extract during the formation of ZnO-NPs from aqueous leaf extract of *M. indica* is depicted in Figure 1. During the formation of ZnO-NPs, zinc ions (Zn^2+^) cap with available phytoconstituents in plant extract to form a complex compound which undergoes direct decomposition during calcination in static air atmosphere finally leading in the formation of ZnO-NPs which is in corroboration with Karnan and Selvakumar [34] and Jafarirad et al. [35]. The biosynthesized ZnO-NPs from *M. indica* were dispersed in sterile distilled water and analyzed by UV-Vis spectral analysis. The results of the spectral analysis read a maximum absorption peak at 359 nm (Figure 2A). The XRD patterns of biosynthesized ZnO-NPs revealed visible peaks at 31.61°, 34.30°, 36.11°, 47.44°, 56.43°, 62.85°, and 67.88° 2θ angles which correspond to (100), (002), (101), (102), (110), (103), and (112) planes, as shown in Figure 2B. The plane values of XRD patterns have aligned with JCPDS No: 79-2205 [36]. The crystalline size of the synthesized nanoparticles calculated by Scherrer’s formula was of ~6–12 nm. The XRD results corroborate with the previous findings of [23] wherein ZnO- NPs were synthesized by *A. vera* leaf extract with a size of 42.8 nm. The FT-IR spectroscopic analysis was carried out to recognize the functional groups in plant extracts, which play a vital role in the processes of reduction and stabilization of the biosynthesis of nanoparticles [37]. The biosynthesized ZnO-NPs bared a peak at 562.54 cm^−1^ which matches to metal oxide (M-O) bond which was absent in plant extract confirming that the extract acted as capping agents during the synthesis of nanoparticles [38,39]. In-plant extract, spectrum bands were observed at 3346.8 cm^−1^ [alcohol/phenol (OH/–H)], 2900 cm^−1^ [alkanes (C–H)], 1631.62 cm^−1^ [primary amine (N–C)], 1344.92 cm^−1^ [symmetric nitro group (N–O)], 1033.43 cm^−1^ [aliphatic amine (C–N)] and 587.15 cm^−1^ [alkyl halides (C–Br)] (Figure 3). The results are in conformity with the outcomes of other investigators, where the absorption peak that was read between 400 to 600 cm^−1^was designated as a metal oxide bond [22,40]. The morphology of the biosynthesized ZnO-NPs is given in Figure 4A. Likewise, the quantitative metal analysis by EDS revealed 92.62% purity of ZnO-NPs (Figure 4B). Additionally, the purity of synthesized ZnO NPs was confirmed by XRD results. Similarly, chemically synthesized and green synthesized ZnO nanoparticles showed the presence of high-quality ZnO-NPs when visualized by SEM and EDS [41,42]. The elemental analysis of ZnO-NPs biosynthesized using *A. vera* leaf extract was also carried out through EDS analysis [23].

### 3.2. Effect of ZnO-NPs on the Formation of Amadori Product

#### Hemoglobin-*δ*-Gluconolactone Assay

The hemoglobin of RBCs reacts with an oxidized analog of glucose viz. *δ*-gluconolactone (*δ*-Glu) resulting in a significant increase in HbA1c levels after incubation. In the present study, ZnO-NPs significantly reduced the formation of HbA1c. In control (only RBCs) HbA1c was found to be negligible but upon glycation (RBCs + *δ-Glu*) HbA1c was higher. However, upon addition of ZnO-NPs (RBCs + *δ-Glu* + different concentrations of ZnO-NPs) formation of HbA1c was significantly inhibited from 1.81% to 26.54% in a dose-dependent manner, while AG (10 µM) (RBCs + *δ-Glu* + AG) inhibited 45.89% of HbA1c formation. A significant increase in percent HbA1c levels was observed when the RBCs of hemoglobin were treated with *δ*-Glu (Appendix A). It has been observed that under hyperglycemic conditions, the formation of HbA1c was found to be increased compared to normal conditions [25]. The results of the study affirm the same. Reduction in HbA1c levels by ZnO-NPs may help to reduce the AGEs related diseases by delaying or by preventing hemoglobin oxidation.

### 3.3. Inhibitory Effect of ZnO-NPs on MGO Mediated Protein Glycation

#### 3.3.1. Effect of ZnO-NPs on Inhibition of AGEs Formation

Albumin is highly prone to non-enzymatic glycation since it contains the most commonly glycated residues, such as arginine and lysine [43,44]. The auto-oxidation of reducing sugars results in the formation of an intermediate compound called methylglyoxal (MGO), a highly reactive di-carbonyl compound [45,46]. The formation of reactive oxygen species (ROS) and oxidative stress with MGO results in increased protein carbonyl formation and thus reduces thiol-containing protein [47]. In the present study, the co-incubation of BSA with MGO and ZnO-NPs resulted in a decrease in fluorescence, indicating the positive effect of nanoparticles in inhibition of total AGEs and argpyrimidine in a dose-dependent manner (Figure 5 and Appendix A). The results are in agreement with our previous study, wherein ZnO-NPs synthesized from aqueous *A. vera* leaf extract possessed an inhibitory effect on AGE formation in a concentration-dependent manner [23]. This inhibitory activity of ZnO-NPs attributes to its carbonyl scavenging potential, and the results of the present study are in line with earlier studies on glucose-treated BSA [48].

#### 3.3.2. Protective Effect of ZnO-NPs on Red Blood Corpuscles

During hyperglycemic condition increases the glucose concentration in RBCs undergoes auto-oxidation by forming an intermediate compound called MGO, which is known as reactive dicarbonyl compound which causes degenerative changes in RBC cells [49,50]. The formation of echinocyte is due to the elevated levels of MGO, which disrupts the normal function RBCs. The results of the present study on RBCs showed typical biconcave shape in the absence of MGO (Figure 6A), while the presence of MGO resulted in complete damage of biconcave RBCs to irregular shape (Figure 6B). The morphological changes induced by MGO were prevented when the RBCs were treated with AG (Figure 6C) and ZnO-NPs (Figure 6D), stating that the ZnO-NPs help to prevent the complications caused by glycation. The structural alteration of RBCs will eventually lead to functional loss, which ultimately causes oxidative stress [7]. The results affirm that the ZnO-NPs can help to prevent oxidative stress during the process of glycation as they are known to possess potent antioxidant properties [22].

### 3.4. Inhibitory Effect of ZnO-NPs on N-Acetylglycyl-Lysine Methyl Ester (G.K.) Peptide Mediated Ribose Glycation

The inhibition of AGEs formation which is related to protein cross-linking was also evaluated by a GK-peptide–ribose assay [51]. Rahbar et al. [52] demonstrated when G.K. peptide co-incubated with ribose increased the protein glycation products which were in line with the results obtained in the present study. Likewise, a decrease in fluorescence intensity was observed upon ZnO-NPs co-incubation with G.K. peptide and ribose thereby indicating the inhibition of specific AGEs formation (Figure 7). In the present study, ZnO-NPs and AG led to the inhibition of 71.74% and 94.24% AGEs formation (Appendix A). Overall, the study provides evidence of the antiglycation properties of biosynthesized ZnO-NPs from *M. indica* leaf extract on AGEs formation under in vitro conditions.

### 3.5. To Assess the Potential of Biosynthesized ZnO-NPs from M. Indica as Protein Glycation Inhibitor in STZ-Induced Diabetic Rats

#### 3.5.1. Biochemical Studies

The mortality rate in STZ- induced rats, was 22.22%, while the mortality rate in Group 2, 3, and 4 (*n* = 6 rats in an individual group) was at 33.33, 16.66, and 16.66%, respectively. The results indicated that the death of the rats may be due to the toxicity of STZ, the suffocation of lymphatic circulation, malnutrition, infection, and changes in body metabolic conditions which are in agreement with the findings of Wang et al. [53]. The STZ-induced diabetic rats showed significant increases in TG, TC, CR, ALT, AST, ALP and BIL, while HDL, TP and ALB levels were decreased significantly compared to normal rats. It was observed that diabetic rats treated with AG and ZnO-NPs showed decreased levels of TG, CL, ALT, AST, ALP, BIL, and Cr, while HDL, TP, and ALB levels were significantly increased after eight weeks compared with that of diabetic rats (untreated) thereby confirming the positive effects of AG and ZnO-NPs towards the control of diabetics (Table 1). Similarly, there are reports on the control of diabetics in STZ-induced rats upon treatment with myrtenal [54] and naringenin [55] to STZ-induced rats. Treatment with AG and ZnO-NPs in STZ-induced diabetic rats effectively reduced the elevated level of serum lipid and improved the levels of liver activities and renal function enzymes, which is correlated with the lowering risk of vascular diseases and potential against hepatotoxicity and renal toxicity, respectively.

#### 3.5.2. Histopathological Studies

The histopathological effect of ZnO-NPs was carried in the kidney and liver of STZ induced diabetic rats. The transverse section of kidney of STZ- induced rats showed severe degeneration of glomeruli, glomerular basement membrane (GBM) thickening, mesangial expansion, and focal necrosis of tubules (Figure 8) which is attributed to the increased diuresis and renal hypertrophy, oxidative stress, stimulation of renin-angiotensin system, and expression of inflammatory markers in kidney which are responsible for the expansion of mesangial matrix [56]. The observed pathological changes were decreased in STZ-induced diabetic rats, which were treated with AG and ZnO-NPs. The ZnO-NPs treated rats (STZ-induced) showed typical re-appearance of damaged glomeruli and tubules by indicating the protective effect of the treatment, which was also similar to the AG-treated STZ- induced rats. The histopathological results of the liver of normal and STZ-induced diabetic rats treated with ZnO-NPs showed flat, irregular plates that are arranged radially (Figure 8). It was also observed that the hepatocellular region of STZ-induced diabetic rats accompanied by hepatocytes apoptosis (dark-stained and shrunken cells) necrosis. Diabetic rats treated with ZnO-NPs showed regeneration of liver cells to normal in the present study and the observed activity may be due to its antioxidants or antidiabetic potentiality. The findings of the current study are in corroboration with Al-Quraishy et al. [57], wherein selenium nanoparticles treatment to streptozotocin-induced diabetic rats offered protective effect and minimized the risk of diabetic complications which is attributed to the antioxidant and antidiabetic properties activities of the nanoparticles.

#### 3.5.3. Molecular Docking of BSA with ZnO-NPs and AG

The interaction of MGO, AG, and ZnO-NPs with BSA was studied by molecular docking tools. It has been reported that the masking of free amino groups might be responsible for the antiglycation effects [58]. The result of the molecular docking studied is presented in Figure 9. It has been proven that arginine and lysine residues are the most common glycation sites in albumin both in vivo and in vitro [59]. Our results also concur with these findings wherein, MGO, AG and ZnO-NP bound with BSA in the same binding site, where electrostatic interaction with Lys and Arg residues are the most important, as shown in Figure 9. The binding energies of ZnO-NPs (−12.6kcal mol^−1^) showed the highest affinity for BSA when compared to the AG (−4.7 kcal mol^−1^). The affinity of ZnO-NPs with BSA was mainly by hydrogen bonding interaction. The residues interacting with the ZnO-NPs with the formation of hydrogen bonds (green dotted lines) are depicted in Figure 9C,D. From the observations of docked structure, it was evident that ZnO-NPs bind with BSA by mainly interacting with arginine 435 and lysine 187 residues. The distance of H-bonds formed by ZnO-NPs with arginine and lysine was 2.5 and 2.4 Å, respectively. Taking into account that the accessibility and reactivity of arginine residues in BSA are probable determinants of reactivity with MGO, we can infer that the ZnO-NPs decrease the accessibility to Arg and Lys residues. Based on the outcome of docking studies, we could hypothesize that masking of lysine and arginine residues is one of the possible mechanisms responsible for the potent antiglycation activity of ZnO-NPs. The results from the molecular interaction studies of ZnO-NPs were in good agreement with the antiglycation activity of aspartic acid and troxerutin, wherein masking of lysine and arginine residues were observed during their protective role towards the glycation of albumin and hemoglobin [7].

## 4. Conclusions

*M. indica* was used for the first time to biosynthesize zinc oxide nanoparticles (ZnO-NPs) from the aqueous leaf extract to evaluate the potential to inhibit AGEs formation. The biosynthesized ZnO-NPs subjected to physico-chemical characterization revealed 92.62% of purity with ~6–12 nm in size. It was evident that ZnO-NPs were able to inhibit AGEs formation and also protect RBCs.Histopathological results revealed the regeneration of liver and renal organs to normal in STZ-induced diabetic rats. The molecular docking studies successfully showed the identity of binding sites of interaction with BSA during inhibition of AGEs by paving the way for the exploitation of ZnO-NPs as potential drug molecules through ligand-based drug design approaches. Therefore, ZnO-NPs may be exploited as a therapeutic agent to maintain glycemic control and delay the onset of diabetic complications. On the other hand, further investigations on clinical trials are warranted to study the effect of ZnO-NPs on different body tissues.

## Figures and Tables

**Figure 1 biomolecules-09-00882-f001:**
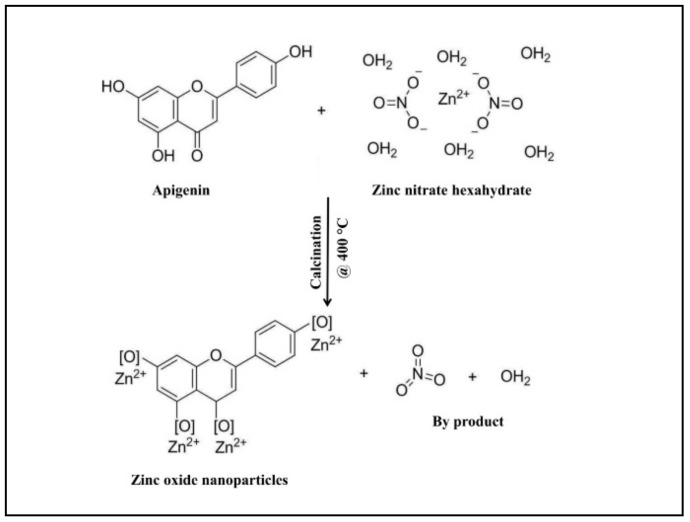
The plausible mechanism of formation of zinc oxide nanoparticles (ZnO-NPs) from *M. indica* aqueous leaf extract.

**Figure 2 biomolecules-09-00882-f002:**
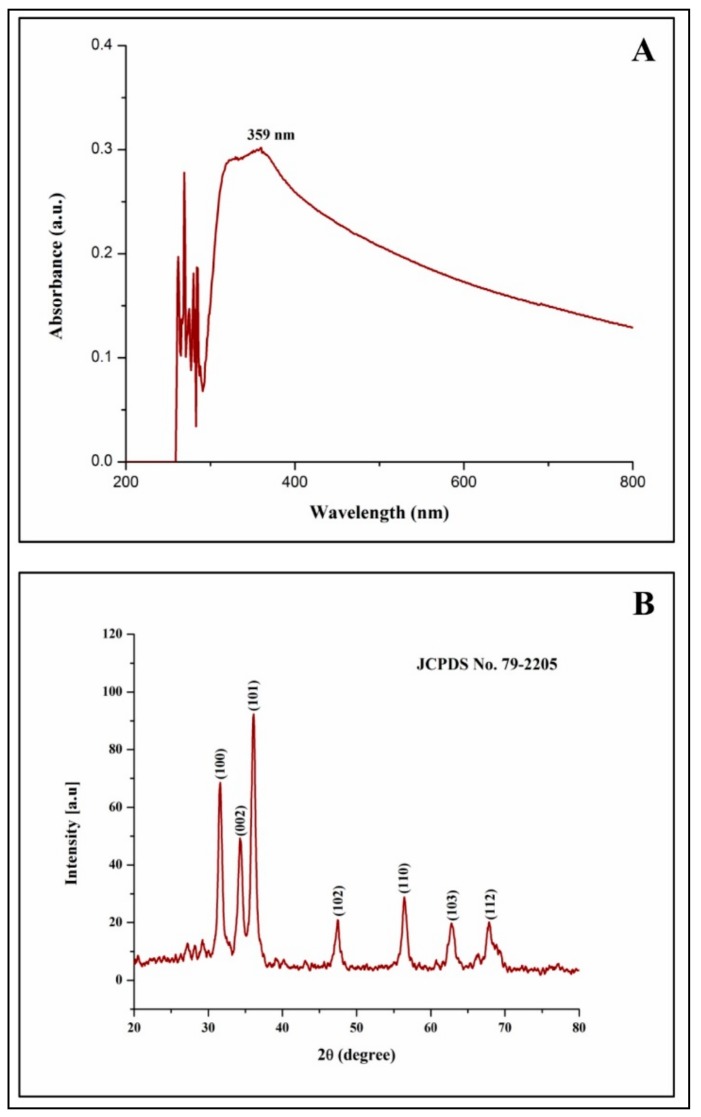
Ultraviolet (UV)-Vis spectroscopy (**A**) and *X*-ray diffraction spectroscopy (**B**) of biosynthesized ZnO-NPs from *M. indica* aqueous leaf extract.

**Figure 3 biomolecules-09-00882-f003:**
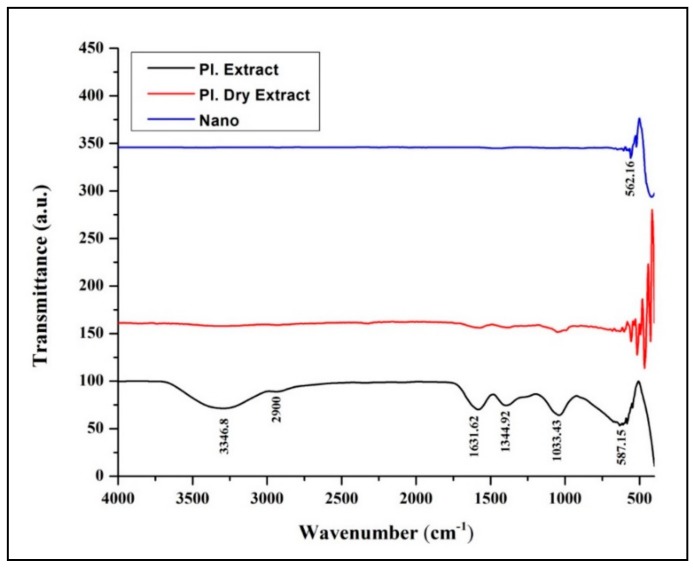
FT-IR analysis of *M. indica* aqueous leaf extract, dry leaf extract, and biosynthesized ZnO-NPs.

**Figure 4 biomolecules-09-00882-f004:**
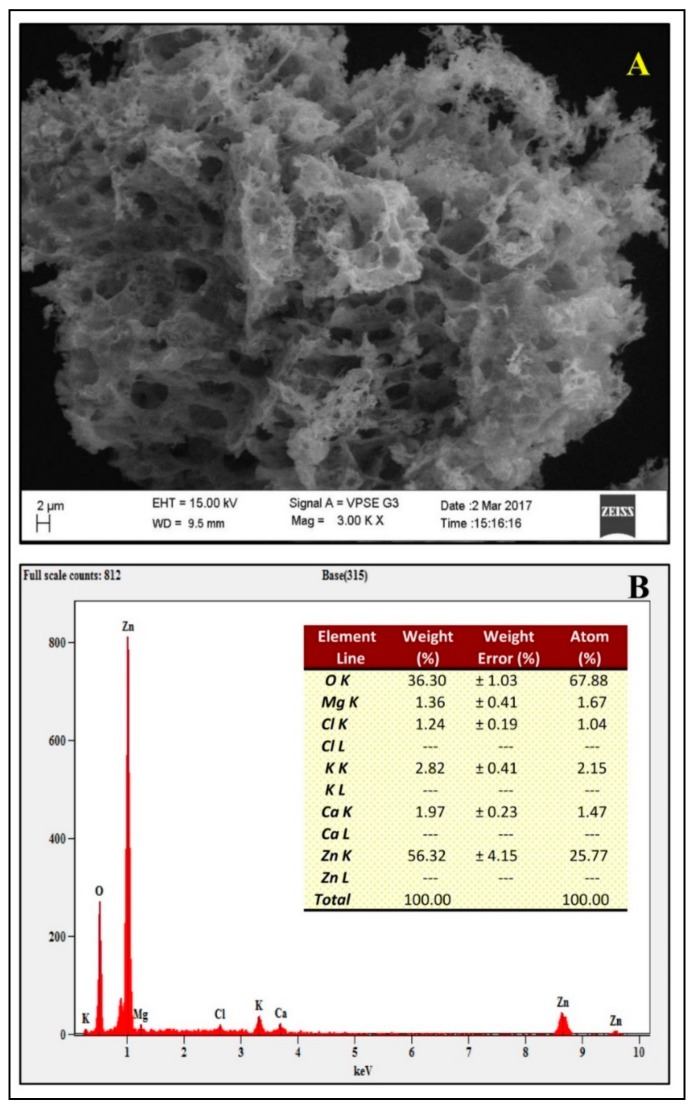
Scanning electron microscopy (**A**) and energy dispersive spectroscopy (**B**) of biosynthesized ZnO-NPs from *M. indica* aqueous leaf extract. EHT- Electron high tension; WD- Working distance

**Figure 5 biomolecules-09-00882-f005:**
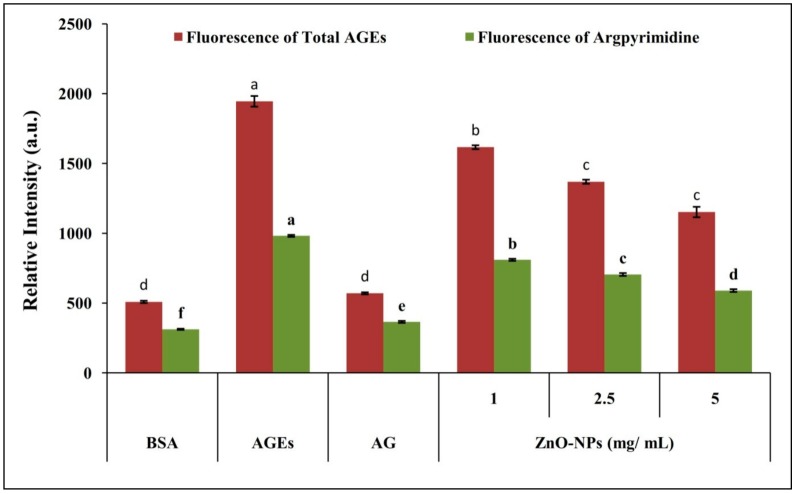
Inhibitory effect of biosynthesized ZnO-NPs from *M. indica* on MGO mediated protein glycation. Each value is the mean of three replicates (*n* = 3), and bars sharing the same letters are not significantly different (*p* < 0.05) according to Tukey’s HSD. The vertical bar indicates the standard error.

**Figure 6 biomolecules-09-00882-f006:**
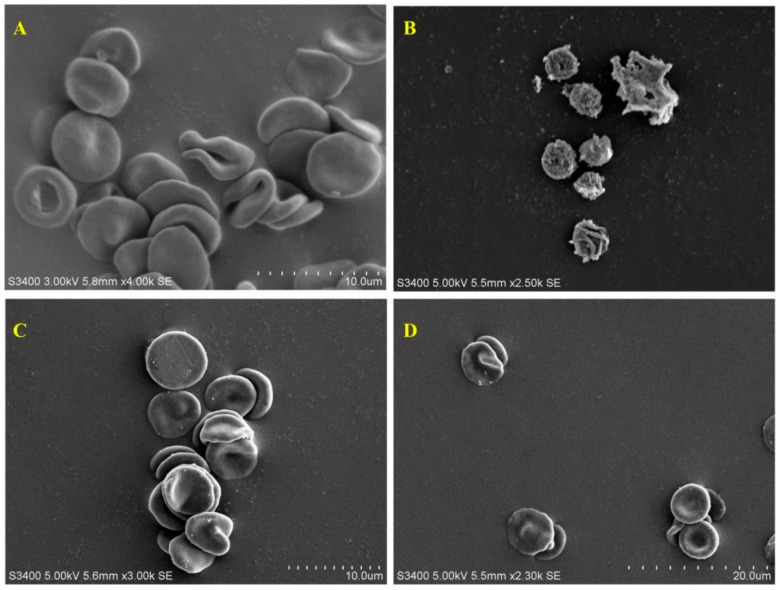
Effect of MGO and ZnO-NPs on red blood corpuscles (RBCs). (**A**) Normal RBCs showing normal shape. (**B**) MGO on RBCs showing the formation of echinocyte. (**C**) MGO on RBCs in the presence of AG. (**D**) MGO on RBCs in the presence of biosynthesized ZnO-NP.

**Figure 7 biomolecules-09-00882-f007:**
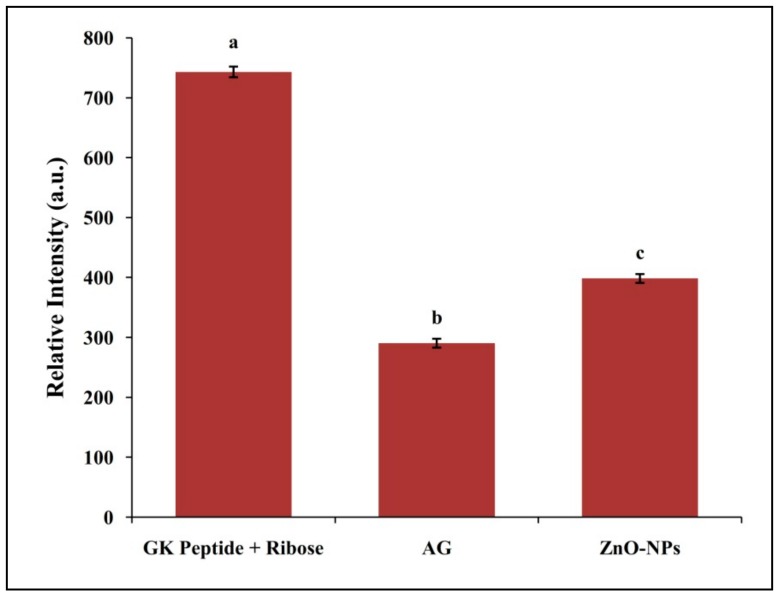
GK-peptide ribose assays. The fluorescence of the GK-ribose mixture with and without biosynthesized ZnO-NPs was recorded at (Ex 340 nm and Em 420 nm). Each value is the mean of three replicates (*n* = 3) and bars sharing the same letters are not significantly different (*p* < 0.05) according to Tukey’s HSD. The vertical bar indicates the standard error.

**Figure 8 biomolecules-09-00882-f008:**
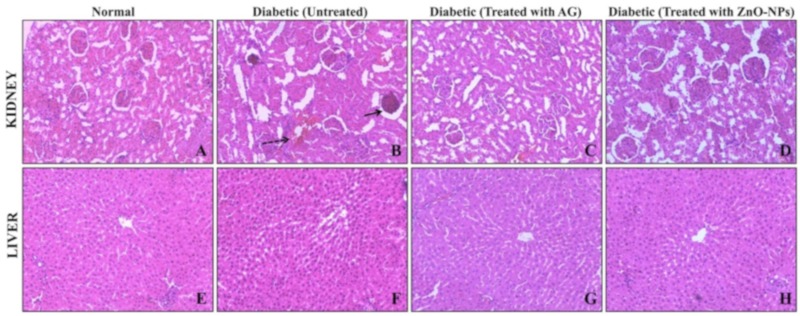
The histopathological effect of ZnO-NPs in STZ-induced diabetic rats was carried in the kidney and liver. (**A**,**E**) are the cross-sections of kidney and liver, respectively, of normal rats. (**B**,**F**) are the cross-sections of kidney and liver, respectively, of STZ-induced diabetic rats, (**C**,**G**) are the cross-sections of kidney and liver, respectively, of STZ-induced diabetic rats treated with aminoguanidine, (**D**,**H**) are the cross-sections of kidney and liver, respectively, of STZ-induced diabetic rats treated with ZnO-NPs.

**Figure 9 biomolecules-09-00882-f009:**
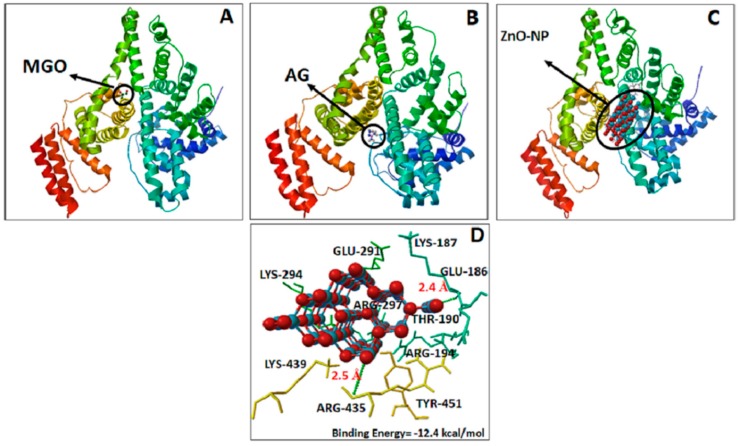
Bovine serum albumin (BSA) complex interaction and their binding sites with methylglyoxal (**A**), aminoguanidine (**B**) and zinc oxide nanoparticles (**C**,**D**). The ligands are presented as ball and stick and receptor in cyan, yellow, and orange. The dashed green lines represent hydrogen-bonding interaction.

**Table 1 biomolecules-09-00882-t001:** Effect of six weeks of treatment of ZnO-NPs on biochemical estimations in STZ-induced diabetic rats.

RATS	Biochemical Analysis of Blood Serum										
	ALT (U L^−1^)	AST (U L^−1^)	ALP (U L^−1^)	CRE (MG DL^−1^)	BUN (MG DL^−1^)	ALB (G DL^−1^)	TP (G DL^−1^)	BIL (MG DL^−1^)	TG (MG DL^−1^)	TC (MG DL^−1^)	HDL (MG DL^−1^)
**Group 1**	86.29 ± 2.90	263.51 ± 5.85	243.06 ± 22.71	0.74 ± 0.045	19.38 ± 1.02	3.52 ± 0.30	7.35 ± 0.41	0.26 ± 0.03	48.53 ± 2.07	55.90 ± 2.14	48.53 ± 2.07
**Group 2**	197.62 ± 6.44 *^,a^	477.11 ± 4.99 *^,a^	823.86 ± 9.56 *^,a^	0.96 ± 0.31 *^,a^	51.50 ± 2.62 *^,a^	1.37 ± 0.24 *^,a^	4.75 ± 0.29 *^,a^	0.38 ± 0.01 *^,a^	143.81 ± 3.44 *^,a^	77.28 ± 2.85 *^,a^	38.41 ± 1.17 *^,a^
**Group 3**	110.90 ± 3.03	275.68 ± 5.42	276.47 ± 7.80	0.75 ± 0.035 ^b^	23.676 ± 1.4 ^b^	2.97 ± 0.17 ^b^	6.73 ± 0.39^b^	0.22 ± 0.03 ^b^	59.21 ± 1.44	58.10 ± 1.27 ^b^	44.50 ± 1.52 ^b^
**Group 4**	125.65 ± 3.08	321.80 ± 5.34	532.87 ± 8.56	0.74 ± 0.02 ^b^	22.06 ± 1.48 ^b^	2.78 ± 0.13 ^b^	6.24 ± 0.13 ^b^	0.21 ± 0.02 ^b^	96.84 ± 1.92	57.39 ± 1.87 ^b^	46.88 ± 1.18 ^b^

Note: Alanine aminotransferase (ALT), aspartate aminotransferase (AST), alkaline phosphatase (ALP), creatinine (CR), blood urea nitrogen (BUN), albumin (ALB), total protein (TP), bilirubin (BIL), triglyceride (TG), serum’s total cholesterol (TC), and high-density lipoprotein (HDL). Values are mean of each group (*n* = 6) and ± indicate standard errors according to Tukey’s multiple comparison test, * *p* < 0.05 (normal vs.diabetic), ^a^
*p* < 0.05 (diabetic vs. AG and ZnO-NPs) and ^b^
*p* > 0.05 (AG vs. ZnO-NPs).Group 1-Healthy control rats, Group 2-STZ-induced hyperglycemic rats without treatment, Group 3-STZ-induced hyperglycemic rats treated with AG, Group 4-STZ-induced hyperglycemic rats treated with ZnO-NPs.

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
