# Peer review of "Biosynthesized ZnO-NPs from Morus indica Attenuates Methylglyoxal-Induced Protein Glycation and RBC Damage: In-Vitro, In-Vivo and Molecular Docking Study"

_biomolecules, 2019, doi:10.3390/biom9120882_

Round 1

Reviewer 1 Report

The manuscript entitled “Biosynthesized ZnO-NPs attenuates methylglyoxal-induced protein glycation and RBC damage: In-vitro, in-vivo and molecular docking study.” by Dr. Satish A. are well written manuscript and results were very clear. The authors demonstrated that the biosynthesized ZnO-NPs made from zinc nitrate and aqueous leaf extract of Morus indica, showed anti-glycative effect in both in vitro and in vivo. The manuscript is really interesting, however, I would like to ask several things.

Q1. To make ZnO-NPs, the authors did not purify the water extract of Morus indica. Do you have any idea which compounds in Morus indica have antiglycation activity?

Q2. I guess the authors called “ZnO-NPs” means “ZnO-NPs with water extract of Morus indica”. And the authors mention just ZnO-NPs (without water extract of Morus indica) did not have any antiglycation activity (lane 144-146). As shown in Figure 8, the ZnO-NPs with Morus indica bound with BSA in arginine 435 and lysine 465. How about ZnO-NPs without Morus indica? This question is connected with Q1. Do you think the efficacy of ZnO-NPs with water extract of Morus indica against glycation is coming from some compounds in Morus indica? Or did some compounds in Morus indica alter structures of ZnO-NPs? 

Q3. In Figure 4, the authors showed only fluorescence pictures. Please show also blight field one.

Minor comments:

In title and line 51, authors wrote “in-vitro”. However in lane 84, there is “in vitro”. It should be same style.

Even though ZnO-NPs already appeared in lane 96, formal name (zinc oxide nanoparticles) was firstly showed in lane 98. Please change it.

In the section of “3.5” and Table 1, the authors did not mention which conditions were Group 1-4. Even though the authors mention in the section of “2.9.2”, I guess they may write down again to better understanding.

In table 1, I am a bit confused the numbers separated in 2 lines. If it is difficult to put in one lane, Please write down such as

“86.29

±2.90”

instead of  

“86.29±2.9

0”

Author Response

REVIEWER 1

The manuscript entitled “Biosynthesized ZnO-NPs attenuates methylglyoxal-induced protein glycation and RBC damage: In-vitro, in-vivo and molecular docking study.” by Dr. Satish A. are well written manuscript and results were very clear. The authors demonstrated that the biosynthesized ZnO-NPs made from zinc nitrate and aqueous leaf extract of Morus indica, showed anti-glycative effect in both in vitro and in vivo. The manuscript is really interesting, however, I would like to ask several things.

Q1. To make ZnO-NPs, the authors did not purify the water extract of Morus indica. Do you have any idea which compounds in Morus indica have antiglycation activity?

Response: In our previous study (Satish et al., 2019) the water extract of Morus indica showed the presence of apigenin, which possesses the anti-glycation property. In our previous studies we also isolated and characterized apigenin from Morus indica and evaluated for its anti-glycation effect. Hence in the revised manuscript the plausible mechanism of ZnO-NPs has been provided (Fig. 1).

Ref: S. Anandan & Urooj, A. (2019). Bioactive Compounds from Morus indica as Inhibitors of Advanced Glycation End Products. Indian Journal of Pharmaceutical Sciences81(2), 282-292.

Q2. I guess the authors called “ZnO-NPs” means “ZnO-NPs with water extract of Morus indica”. And the authors mention just ZnO-NPs (without water extract of Morus indica) did not have any antiglycation activity (lane 144-146). As shown in Figure 8, the ZnO-NPs with Morus indica bound with BSA in arginine 435 and lysine 465. How about ZnO-NPs without Morus indica? This question is connected with Q1. Do you think the efficacy of ZnO-NPs with water extract of Morus indica against glycation is coming from some compounds in Morus indica? Or did some compounds in Morus indica alter structures of ZnO-NPs? 

Response:

(i) I guess the authors called “ZnO-NPs” means “ZnO-NPs with water extract of Morus indica”. And the authors mention just ZnO-NPs (without water extract of Morus indica) did not have any antiglycation activity (lane 144-146). As shown in Figure 8, the ZnO-NPs with Morus indica bound with BSA in arginine 435 and lysine 465. How about ZnO-NPs without Morus indica?

Response: The differences between them are related to sizes and surfaces. ZnO nanopowder has a size of nanoparticle <100 nm while the synthesized ZnO nanoparticles from M. indica have a size of ~6-12 nm. Nanoparticle can progressively and selectively adsorb biomolecules when they come into contact with complex biological fluids (M. indica in the present research). This surface, called ‘corona’ of biomolecules lowers the surface energy of the nanoparticle and promotes its dispersion and higher interaction with the biological system. According to the bibliography, these interactions lead to the formation of the protein corona, particle wrapping, intracellular uptake and biocatalytic process (NATURE NANOTECHNOLOGY, Vol 7, 2012, and DOI: 10.1038/NNANO.2012.207). On the other hand, computational simulations, though a very complicated task for the macromolecular assembly onto a ZnO-NPs surface, can complement the experimental research in an effective manner. Thus, a flat ZnO-NPs surface (15 Å x 25 Å) mimicking the particle curvature of ZnO-NP was constructed, for that reason, we choose a nanoparticle surface to evaluate the interaction using docking molecular.

Ref: Monopoli, M. P., Åberg, C., Salvati, A., & Dawson, K. A. (2012). Biomolecular coronas provide the biological identity of nanosized materials. Nature nanotechnology, 7(12), 779.

(ii) Do you think the efficacy of ZnO-NPs with water extract of Morus indica against glycation is coming from some compounds in Morus indica? Or did some compounds in Morus indica alter structures of ZnO-NPs? 

Response: The compounds present in Morus indica can alter the structure of ZnO-NPs formed which will affect the biological nature of the particles. In the revised manuscript, we have provided a plausible mechanism (Fig. 1) during the synthesis of ZnO-NPs.

Q3. In Figure 4, the authors showed only fluorescence pictures. Please show also blight field one.

Response: We thank the reviewer for their valuable suggestions. The facility of bright field microscope was not available in the University and hence, we experimented using fluorescence microscopy. We make sure that the advice will be considered in our future studies.

Minor Comments:

Q1. In title and line 51, authors wrote “in-vitro”. However in lane 84, there is “in vitro”. It should be same style.

Response: We thank the reviewer for the keen observation and as per the suggestion, uniformity in the mentioning of in-vitro has been maintained throughout the revised manuscript.

Q2. Even though ZnO-NPs already appeared in lane 96, formal name (zinc oxide nanoparticles) was firstly showed in lane 98. Please change it.

Response: As per the suggestion, the formal name has been first represented into full form in Line No. 48 of the revised manuscript and after that, it has been abbreviated throughout the manuscript.

Q3. In the section of “3.5” and Table 1, the authors did not mention which conditions were Group 1-4. Even though the authors mention in the section of “2.9.2”, I guess they may write down again to better understanding.

Response: As per the reviewer's suggestion, for better understanding, the treatment name has been changed to Group 1 to Group 4 and the conditions used for each group are given as a footnote to Table 1.

Q4. In table 1, I am a bit confused the numbers separated in 2 lines. If it is difficult to put in one lane, Please write down such as“86.29±2.90”instead of  “86.29±2.90”

Response: As per the reviewer's suggestion, the mistake has been rectified in the revised manuscript to avoid confusion.

Reviewer 2 Report

Major comments:

Above titled manuscript is carefully written with good English, organization and illustrations. In this work, the authors studied inhibition activity of biosynthesized ZnO-NPs against protein glycation and RBC damage both in vitro and in vivo. The authors found that plant derived NPs are showing effective and dose dependent anti-glycation effect compared to those obtained with merely chemical way (commercial NPs). In addition, their molecular docking study illustrated that strong efficacy of biosythesied ZnO-NPs achieved with masking methods. In other words, the study is enabling cost effective therapeutic method for diabetes related complications. However, following comments should be highly considered to meet criteria of the journal.      

Secondary comments:

What is benefit for synthesizing ZnO – NPs from Morus indica L. compared to other plants which used with same purpose? Which functional groups/compounds of this plant are improving the crystallite and nano size of the particle?

Please include analysis result for heat processed plant without adding zinc nitrate. It should be proven that peak at 562.54 cm-1 is exactly match with ZnO-NPs’ peak. In other words, trace amount (under detection limit of FTIR in leaf extract) of heavy metals can be accumulated in Morus indica L. in nature and their concentration can be increased and read at FT-IR instrument after procedure in 2.3.

The author said that SEM characterization was used to check aggregation. But there is no explanation about the aggregation. From the SEM image, the particles seem like just one big particle (not nano). There is no information about the size and shape of the particles. The confirmation of ZnO-NPs should be informed by EDX mapping or XPS as well.    

The reason for decreases (reference – 400OC and manuscript 300OC is applied) in calcination temperature for producing ZnO nanoparticle should be explained.

Based on which reference, 30 mg/kg dose of AG is selected for treatment and incidence of induction streptozotocin of diabetes in rats? As well as all other references should be included in section 2.9.2.

Author Response

REVIEWER 2

Above titled manuscript is carefully written with good English, organization and illustrations. In this work, the authors studied inhibition activity of biosynthesized ZnO-NPs against protein glycation and RBC damage both in vitro and in vivo. The authors found that plant derived NPs are showing effective and dose dependent anti-glycation effect compared to those obtained with merely chemical way (commercial NPs). In addition, their molecular docking study illustrated that strong efficacy of biosythesied ZnO-NPs achieved with masking methods. In other words, the study is enabling cost effective therapeutic method for diabetes related complications. However, following comments should be highly considered to meet criteria of the journal.      

Q1. What is benefit for synthesizing ZnO – NPs from Morus indica L. compared to other plants which used with same purpose? Which functional groups/compounds of this plant are improving the crystallite and nano size of the particle?

Response: Morus indica was selected based on the previous studies from our research group (Satish et al., 2019) wherein we could find strong inhibition of AGEs formation upon treatment with plant extract and was attributed towards the presence of flavonoids (apigenin and luteolin). We also state that the flavonoid compounds may have participated in the formation of nanoparticles. The plausible mechanism of formation of ZnO-NPs is given in the revised manuscript.

Ref: S. Anandan & Urooj, A. (2019). Bioactive Compounds from Morus indica as Inhibitors of Advanced Glycation End Products. Indian Journal of Pharmaceutical Sciences81(2), 282-292.

Q2. Please include analysis result for heat processed plant without adding zinc nitrate. It should be proven that peak at 562.54 cm-1 is exactly match with ZnO-NPs’ peak. In other words, trace amount (under detection limit of FTIR in leaf extract) of heavy metals can be accumulated in Morus indica L. in nature and their concentration can be increased and read at FT-IR instrument after procedure in 2.3.

 Response: As per the reviewer’s suggestion, the FT-IR of the heat-processed plant extract has been provided in the revised manuscript (Fig. 3) and we can’t see any prominent peaks at the spectral range thereby indicating the secondary metabolites may have capped during the synthesis of ZnO-NPs. It may be noted that the FT-IR analysis is carried out in order to know the efficacy of the secondary metabolites present in plant extract has interacted during the synthesis of ZnO-NPs.

Q3. The author said that SEM characterization was used to check aggregation. But there is no explanation about the aggregation. From the SEM image, the particles seem like just one big particle (not nano). There is no information about the size and shape of the particles. The confirmation of ZnO-NPs should be informed by EDX mapping or XPS as well.    

Response: As per the reviewer’s suggestion, in Line No. 308-309 the term “aggregation” has been deleted and the sentence has been rephrased accordingly.

Q4. The reason for decreases (reference – 400°C and manuscript 300°C is applied) in calcination temperature for producing ZnO nanoparticle should be explained.

Response: The typographical error has been rectified in the revised manuscript.

Q5. Based on which reference, 30 mg/ kg dose of AG is selected for treatment and incidence of induction streptozotocin of diabetes in rats? As well as all other references should be included in section 2.9.2.

Response: In the revised manuscript, reference for the use of dosage has been added. In the cited reference (Swamy et al., 1996), 25 mg/kg dose of AG was used, but in the present study the dosage of AG was slightly modified (to 30 mg/kg) from the results of our previous studies wherein the usage of AG at 30 mg/kg in STZ induced rats were able to significantly decrease the level of lipid peroxides and also lowered the levels of AGE fluorescence (unpublished data). Hence the same concentration was used in the present study.

Ref: Swamy-Mruthinti, S., Green, K., & Abraham, E. C. (1996). Inhibition of cataracts in moderately diabetic rats by aminoguanidine. Experimental eye research, 62(5), 505-510.

Reviewer 3 Report

The manuscript of A et al. aims to describe how biosynthesized ZnO-NPs are able to inhibit protein glycation. They have performed a broad set of experiments ongoing from computational studies to in vivo studies. Although their findings could hold certain potential, some the interpretations of the obtained results are doubtful and needs a strong revision. In addition, some issues (that I report in the following paragraphs) needs to be properly addressed and/or clarify before this manuscript ca be recommended for publication.

Major issues

-The English of the entire manuscript is really poor. In fact, this reviewer had problems to understand what the authors wanted to say in some paragraphs. Consequently, the manuscript needs to be reviewed by a native English speaker before it can be accepted for publication.

-Most of the sentences in the introduction lack of a logical connection between them. Just as an example (but not the only). In line 84, the authors claim that an AGE inhibitor has been evaluated as inhibitor of diabetes, but it is not clear at all which inhibitor. Moreover, in line 88, they say that “this AGEs inhibitor lacks the health concerns by showing adverse effect [9]”. If we go ref [9] it refers to quercetin, which has not been mentioned along the introduction…. I think that the entire introduction needs a strong revision to make it clearer.

-The inhibitory effect of ZnO-NPs on the formation of AGEs was already proved by Ashraf et al. in 2018 (Mol Neurobiol. 2018 Sep;55(9):7438-7452). Hence, the authors must explain in their rebuttal letter (but also in the introduction) what is the novelty of their manuscript in comparison with the data reported by Ashraf et al. In addition, the authors must deeply introduce this manuscript in the introduction and in the results and discussion section through the comparison of their findings with those from Ashraf et al. Moreover, the introduction needs to include more information on the role of nanoparticles on the glycation process. For instance, it should ideally introduce the findings of manuscripts such as: i) Nanoscale, 2019,11, 13126-13138; ii) Int J Nanomedicine. 2014; 9: 5461–5469; or iii) Nanotechnology 26(14):145703.

-In Lines 106-107, the authors state that “there are no reports on the inhibitory effect of ZnO-NPs biosynthtesized from M. indica on protein glycation”. However, the word carried out by Ashraf et al. (Mol Neurobiol. 2018 Sep;55(9):7438-7452) proved that ZnO-NPs biosynthtesized from Aloe Vera on protein glycation. I found necessary that the authors soften their sentence saying that it is not the first time that biosynthesized ZnO-NPs are able to inhibit glycation.

-The authors should provide experimental data (either in the manuscript or as supplementary information) proving that commercial ZnO-NPs did not display any inhibitory effect on the glycation process. Since they performed these experiments (line 144) they can provide this information easily.

-The authors have used gluconolactone to study the early stage of glycation. The authors should justify its use since this reviewer did not know that it could have potential as glycating compound in vivo. Why the authors did not used glucose?

-Moreover, the authors used gluconolactone to study the early stages of glycation, but they incubated the reaction mixture for 16h. Do they expect that the Schiff base or the Amadori compound is still formed? Do they think that AGEs are not formed yet? The authors should explain what they were expected from the use of gluconolactone after 16 days of incubation.

-In addition, they used MGO to study the intermediate states of glycation. Why? MG reacts rapidly with the protein side chains by forming AGEs…. What they wanted to study with the “intermediate stage”?

-They also used ribose to study the late stage of glycation. Why? What they wanted to study with the “late stage”?  Ribose is a highly reactive carbohydrate with high glycation potential…. I do not understand what is the meaning of “late stage glycation” Please justify.  

-The authors have used a non-physiological concentration of Ribose (0.8M; line 202). They should explain why they selected this high concentration, since the results arising from its use cannot be extrapolated to a physiological context.

-It is almost impossible to interpret the figures displayed in the different panels shown in Figure 1. Labels located at the maximum of the peaks are impossible to be read, nor the labels of the axis. The authors must correct this since the reader should be able to extract their own conclusions from the reported results.

-In Figure 2 the authors have determined the % of HbA1c. To determine the % of inhibition I would have expected that the HbA1c formation on RBCs upon incubation with gluconolactone would have been taken as 100%. However, it is only taken as 14%. Which is the meaning of thes 14%? Please, explain better the figure in the ms. In any case the use of 5mg/ml of ZnO-NPs only induced an inhibiton of the 28% (taking the values of the graph) do you think this is relevant?

-From the data reported in Figure 3A, the authors claim that ZnO-NPs are able to quench the fluorescence intensity of the AGEs (line 326-327). The authors should clarify whether the data shown in Figure 3A arises from a reduction in the formation of AGEs or from a simply quenching effect. Hence, to prove that ZnO-NPs are able to inhibit AGEs formation, the authors should titrate a solution containing Ac-Lys and Ac-Arg (in the same molar rations present in BSA), which was previously glycated with MG, with ZnO-NPs. If they do not see quenching, they can conclude that ZnO-NPs inhbit AGEs formation. If they see quenching, the fluorescence data of Figure 3A is confusing and needs to be removed.

-In line 329-330, the authors claim that ZnO-NPs can scavenger carbonyl compounds. They should explain the mechanism through which ZnO-NPs can do this, as this reviewer questions whether a cation containing compound can acts as nucleophile. If they are not able to do it, they should remove the sentence.

-All the Tht data and the conclusions arising from their use must be removed (the Figure 3B and 4). The excitation spectrum of ThT and the emission spectrum of fluorescent AGEs easily overlap. Therefore, this must interfere with the obtained data. In any case, this data it is not essential for the manuscript.

-In line 444-446, the authors compare their findings with the protective effect of the virgin olive oil. Now, the authors must explain the relation between their ZnO-NPs and the virgin olive oil, as this reviewer could not find any connection between them.

-I do not understand why the authors studied the docking of MG and AG on BSA. It is well known that AG inhibits protein glycation competing with the protein nucleophilic side chains for the reactive dicarbonyls. Hence, there is not any logical reason to study its binding to BSA. Moreover, I do not see any logic in studying the docking between MG and BSA as MG covalently reacts with Arg and Lys side chains, and the scenario of being bounded to BSA through intermolecular interactions its highly unlikely to occur. Hence, the authors should remove the docking data involving AG and MG as it does not have any sense.

-In other hand, I think that the docking data arising from the use of ZnO-NPs is good and reports that a plausible mechanism of inhibition is the protection of Lys and Arg side chains. Nevertheless, the authors should clarify in the manuscript: i) how they get the 3D structure of the ZnO-NPs that they used for the docking; and ii) which is the difference of between this and the structure of the chemically synthetized ZnO-NPs, which are not able to inhibit protein glycation. Would it be true that if the BSA-ZnO-NPs binding is the main mechanism of inhibition, the chemically synthetized ZnO-NPs should also be able to inhibit glycation?

Minor concerns

-Line 47. The authors assume that AGEs formation induce conformational changes on the protein structure. However, recent reports have proven that this does not necessarily occur (BMC Biochem. 2011 Aug 5;12:41; Biomacromolecules. 2014 Sep 8;15(9):3449-62). Hence, I request the authors to replace “conformational changes of the biomolecules” for “the loss of the biomolecular function”.

-Line 48. The authors should define in the abstract what is ZnO-NPs, as it appears there for the first time.

-Lline 58 and 118. “Methyglyoxal” should not be in capitals nor “Bovine”. Please correct.

-Line 59. The inhibition of ZnO-NPs on the glycation-induced conformational changes is not obvious at all, basically because it is not clear that glycation induced conformational changes. Please remove “glycation-induced conformational changes” and leave “glycation-induced aggregation”.

-Line 62. Please define RBCs in the abstract.

-Line 73. AGEs are not only formed under diabetic conditions, but also linked to aging. Please, rephrase this sentence as it gives the idea that AGEs are only linked to diabetes mellitus.

-Line 75. Amino groups do not induce the sugar oxidation. Maillard reaction involves the reaction between reducing sugars and amino acid side chains. Hence, change “gets oxidized” by “reacts”.

-Line 78. What the authors wanted to say with “the process of AGEs”? Would have been better to write “AGEs leads to….”. In addition, I suggest the authors to remove “structural” or change it by “chemical modifications”.

-Line 80. The authors should include a reference proving that AGEs formation induce the formation of disulphur linkage. I never heard about it. If they are not able to provide such a reference, they must remove this sentence.

-Line 81. Why b proteins? The authors must remove “beta” and leave proteins, in general.

-Line 83. When the authors explain the AGEs inhibitors. Why they only focused in AG? There are other inhibitors with higher pharmacological potential, like pyridoxamine. The authors should slightly expand this and cover the most known AGEs inhbitors.

-Line 91. Replace “normal structure of macro molecules” by “native structure of biomolecules”.

-Line 96-97. The authors claim that biosynthesized ZnO-NPs have better antimicrobial activity than chemically synthetized ZnO-NPs. However, the authors did not provide any reference proving it. Hence they should include one or remove the sentence.

-Line 126. The authors must better describe the experimental procedure that they followed to extract the components of the leafs. They should report the temperature, the time, if it was done under agitation, etc…

-Line 140. The authors should remove “under in-vitro” since it is understood from the section itself, but also because in the first sentence they say that they used aminoguanidine “under in-vitro and in-vivo”. 

-Line 144. This last sentence is part of the results, no part of the methodology. Please, move it to the result section.

-Line 219. The authors state that blood glucose levels < 250mg/ml was taken as hyperglycaemic conditions. Is that correct? Should it be > 250mg/ml?

-Line 266. To be statistically significant, p should be < 0.05 no ≤ 0.05. Please correct.

-Line 272. Change “subjected to UV-Vis spectral analysis” by “analyzed by using UV-Vis spectroscopy”.

-Line 277. Remove “as”. In addition, the authors should include the Scherrer’s formula, which would allow the reader to understand how they obtained the size.

-Line 303. Would it be “the hemoglobin of RBCs” instead of “The RBCs of hemoglobin”?

Author Response

REVIEWER 3

The manuscript of A et al. aims to describe how biosynthesized ZnO-NPs are able to inhibit protein glycation. They have performed a broad set of experiments ongoing from computational studies to in vivo studies. Although their findings could hold certain potential, some the interpretations of the obtained results are doubtful and needs a strong revision. In addition, some issues (that I report in the following paragraphs) needs to be properly addressed and/or clarify before this manuscript ca be recommended for publication.

Major Issues

Q1. The English of the entire manuscript is really poor. In fact, this reviewer had problems to understand what the authors wanted to say in some paragraphs. Consequently, the manuscript needs to be reviewed by a native English speaker before it can be accepted for publication.

Response: As per the reviewer’s suggestion, the revised manuscript has been extensively reviewed by an English native speaker and all the correction has been incorporated accordingly.

Q2. Most of the sentences in the introduction lack of a logical connection between them. Just as an example (but not the only). In line 84, the authors claim that an AGE inhibitor has been evaluated as inhibitor of diabetes, but it is not clear at all which inhibitor. Moreover, in line 88, they say that “this AGEs inhibitor lacks the health concerns by showing adverse effect [9]”. If we go ref [9] it refers to quercetin, which has not been mentioned along the introduction…. I think that the entire introduction needs a strong revision to make it clearer.

Response: As per the reviewer’s suggestion, from line 82 to 88 as been rephrased and relevant references have been cited in the revised manuscript. The quoting of reference [9] has been deleted for a better understanding of the sentences in the revised manuscript.

Q3.  (i) The inhibitory effect of ZnO-NPs on the formation of AGEs was already proved by Ashraf et al. in 2018 (MolNeurobiol. 2018 Sep;55(9):7438-7452). Hence, the authors must explain in their rebuttal letter (but also in the introduction) what is the novelty of their manuscript in comparison with the data reported by Ashraf et al.

Response: The novelty of the present work with that of Ashraf et al., has been given in the introduction part in the revised manuscript. Here the authors would like to highlight that, each of the plant extracts will possess its own phytochemical constituents and it differs from plant to plant and hence it also plays a vital role in the activity. Further, the size of the nanoparticle in the present study was between 6-12 nm whereas in Ashraf et al., the size of the particles was 42.8 nm which also plays a vital role in the interaction. In Ashraf et al., the glycation reaction system included IgG as a protein and MGO as a glycating agent which significantly differs from the present study.

(ii) In addition, the authors must deeply introduce this manuscript in the introduction and in the results and discussion section through the comparison of their findings with those from Ashraf et al. Moreover, the introduction needs to include more information on the role of nanoparticles on the glycation process. For instance, it should ideally introduce the findings of manuscripts such as: i) Nanoscale, 2019,11, 13126-13138; ii) Int J Nanomedicine. 2014; 9: 5461–5469; or iii) Nanotechnology 26(14):145703.

Response: The introduction of the revised manuscript has been substantially rewritten with all the above points raised by the reviewer and also compared the results obtained in the present study with our previous studies (Ashraf et al.). We have also cited the refrence which the reviewer has quoted appropriately in the revised manuscript.

Q4. In Lines 106-107, the authors state that “there are no reports on the inhibitory effect of ZnO-NPs biosynthtesized from M. indica on protein glycation”. However, the word carried out by Ashraf et al. (MolNeurobiol. 2018 Sep;55(9):7438-7452) proved that ZnO-NPs biosynthtesized from Aloe Vera on protein glycation. I found necessary that the authors soften their sentence saying that it is not the first time that biosynthesized ZnO-NPs are able to inhibit glycation

Response: As per the reviewer’s suggestion, the sentence in Line 106-107 (now Line 123-124) has been rewritten in the revised manuscript accordingly.

Q5. The authors should provide experimental data (either in the manuscript or as supplementary information) proving that commercial ZnO-NPs did not display any inhibitory effect on the glycation process. Since they performed these experiments (line 144) they can provide this information easily.

Response: As per the reviewer’s suggestion, the data on commercial zinc oxide nanopowder on the inhibitory effect on the glycation process has been provided as Supplementary Table 1,2 and 3.

Q6. The authors have used gluconolactone to study the early stage of glycation. The authors should justify its use since this reviewer did not know that it could have potential as glycating compound in vivo. Why the authors did not used glucose?

Response: We agree with the comments raised by the reviewer. Here we would like to justify that, even though glucose has the ability to increase glycation in human hemoglobin upon interaction but it takes several days to weeks, while δ-Gluconolactone reacts strongly with hemoglobin within hours (Ref: Lindsay et al. 1997) hence it was used in the early stage of glycation. Due to its early reaction to cause glycation compared to glucose, δ-Gluconolactone was used throughout the study.

Ref: Lindsay, R. M., Smith, W., Lee, W. K., Dominiczak, M. H., & Baird, J. D. (1997). The effect of δ-gluconolactone, an oxidised analogue of glucose, on the nonenzymatic glycation of human and rat haemoglobin.Clinicachimicaacta, 263(2), 239-247.

Q7. Moreover, the authors used gluconolactone to study the early stages of glycation, but they incubated the reaction mixture for 16h. Do they expect that the Schiff base or the Amadori compound is still formed? Do they think that AGEs are not formed yet? The authors should explain what they were expected from the use of gluconolactone after 16 days of incubation.

Response: We hereby state that δ-Gluconolactone was used to obtain the Amadori products and it will be formed with 16 h of incubation and it has been proved previously (Ref: Wu et al., 2005). We also confirm that after 16 h incubation there will be no formation of AGEs yet. We just expected the formation of Amadori products after 16 h of incubation.

Ref: Wu, C. H., & Yen, G. C. (2005). Inhibitory effect of naturally occurring flavonoids on the formation of advanced glycation endproducts. Journal of agricultural and food chemistry, 53(8), 3167-3173.

Q8. In addition, they used MGO to study the intermediate states of glycation. Why? MG reacts rapidly with the protein side chains by forming AGEs…. What they wanted to study with the “intermediate stage”?

Response: Here we agree with comments raised by the reviewer and accept that MG reacts rapidly with the amino groups of proteins to form inter and intra-molecular cross-links resulting in the formation of AGEs. The intermediate stage of glycation was carried out to know the ability of ZnO-NPs in the inhibition of glycation induced oxidative stress with MGO. The same has been proved with the results presented in Fig. 5 and 6.

Q9. They also used ribose to study the late stage of glycation. Why? What they wanted to study with the “late stage”?  Ribose is a highly reactive carbohydrate with high glycation potential…. I do not understand what is the meaning of “late stage glycation” Please justify.  

Response: We hereby state that the term “late stage of glycation” was used because here irreversible compounds (AGEs) will be formed. As you have only stated, ribose has a highly reactive carbohydrate resulting in high glycation potential to form AGEs i.e. lysine residue present in the G.K. peptide is able to generate peptides with advanced Maillard reaction product with dimerization through lysine-lysine cross-linking. The co-incubation of G.K. peptide with ribose increases late glycation product formation and hence this model system was used to evaluate the inhibitory effects of ZnO-NPs on protein cross-linking.

Q10. The authors have used a non-physiological concentration of Ribose (0.8M; line 202). They should explain why they selected this high concentration, since the results arising from its use cannot be extrapolated to a physiological context.

Response: The concentration of Ribose (0.8 M) was used based on the earlier studies reported by Wu et al., (2005) and the physiological effect of the ribose concentration is not yet elucidated according to our knowledge.

Ref: Wu, C. H., & Yen, G. C. (2005). Inhibitory effect of naturally occurring flavonoids on the formation of advanced glycation endproducts. Journal of agricultural and food chemistry, 53(8), 3167-3173.

Q11. It is almost impossible to interpret the figures displayed in the different panels shown in Figure 1. Labels located at the maximum of the peaks are impossible to be read, nor the labels of the axis. The authors must correct this since the reader should be able to extract their own conclusions from the reported results.

Response: As per the reviewer’s suggestion, each of the figures displayed in different panels in Fig. 1 have been separated individually in the revised manuscript.

Q12. In Figure 2 the authors have determined the % of HbA1c. To determine the % of inhibition I would have expected that the HbA1c formation on RBCs upon incubation with gluconolactone would have been taken as 100%. However, it is only taken as 14%. Which is the meaning of thes 14%? Please, explain better the figure in the ms. In any case the use of 5mg/ml of ZnO-NPs only induced an inhibiton of the 28% (taking the values of the graph) do you think this is relevant?

Response: As per the reviewer’s suggestion, the values depicted in the graph were calculated based on the formula provided in column 1 of the below table. Further, the data representation has been changed as per the suggestion of the reviewer, it has been given has Suppl. Table 1 in the revised manuscript.

COLUMN 1

COLUMN 2

Samples

Control ( normal Hb)

6.01 ± 0.08e

-

Hb + δ-Gluconolactone

14.35 ± 0.13a

-

Hb + δ-Gluconolactone + AG

7.76 ± 0.16d

45.89 ±  1.14a

Hb+ δ-Gluconolactone + ZnO-NPs

1 mg

14.08 ± 0.12a

1.81 ± 0.87d

2.5 mg

12.34 ± 0.18b

13.97± 1.30c

5 mg

10.54 ± 0.17c

26.54 ± 1.21b

Hb+ δ-Gluconolactone + ZON

1 mg

14.31 ± 0.12a

0.27 ± 0.48d

2.5 mg

14.27 ± 0.12a

0.53 ± 0.86d

5 mg

14.18 ± 0.14a

1.13 ± 0.97d

Where

            AHbA1c is Absorbance of HbA1c against distilled water

AHbTOTAL is Absorbance of HbA1c against distilled water

VHbA1c is Volume of HbA1c against distilled water

VHbTOTAL is Volume of HbA1c against distilled water

The absorbance of Hb + δ-Gluconolactone The absorbance of Hb + δ-Gluconolactone + samples tested

Q13. From the data reported in Figure 3A, the authors claim that ZnO-NPs are able to quench the fluorescence intensity of the AGEs (line 326-327). The authors should clarify whether the data shown in Figure 3A arises from a reduction in the formation of AGEs or from a simply quenching effect. Hence, to prove that ZnO-NPs are able to inhibit AGEs formation, the authors should titrate a solution containing Ac-Lys and Ac-Arg (in the same molar rations present in BSA), which was previously glycated with MG, with ZnO-NPs. If they do not see quenching, they can conclude that ZnO-NPs inhibit AGEs formation. If they see quenching, the fluorescence data of Figure 3A is confusing and needs to be removed.

Response: Thank you for your valuable idea regarding proving ZnO-NPs were able to inhibit the AGEs formation. The confusion regarding the terminology “quenching” has been removed and the sentence has been reframed accordingly in the revised manuscript.

Further, in the present study the data provided is from a reduction in the formation of AGEs and not the quenching effect where MGO readily reacts with lysine and arginine residues of protein to produce non-enzymatic protein glycation and subsequent formation of AGEs, which was measured by excitation (330 nm) and emission (440 nm).

Q14. In line 329-330, the authors claim that ZnO-NPs can scavenger carbonyl compounds. They should explain the mechanism through which ZnO-NPs can do this, as this reviewer questions whether a cation containing compound can acts as nucleophile. If they are not able to do it, they should remove the sentence.

Response: As per the reviewer’s suggestion, the sentence has been removed in the revised manuscript.

Q15. All the Tht data and the conclusions arising from their use must be removed (the Figure 3B and 4). The excitation spectrum of ThT and the emission spectrum of fluorescent AGEs easily overlap. Therefore, this must interfere with the obtained data. In any case, this data it is not essential for the manuscript.

Response: As per the reviewer’s suggestion, the Tht data and its conclusions have been removed in the revised manuscript.

Q16. In line 444-446, the authors compare their findings with the protective effect of the virgin olive oil. Now, the authors must explain the relation between their ZnO-NPs and the virgin olive oil, as this reviewer could not find any connection between them.

Response: As per the reviewer’s suggestion, the sentence has been modified in the revised manuscript (Line 446-450).

Q17. I do not understand why the authors studied the docking of MG and AG on BSA. It is well known that AG inhibits protein glycation competing with the protein nucleophilic side chains for the reactive dicarbonyls. Hence, there is not any logical reason to study its binding to BSA. Moreover, I do not see any logic in studying the docking between MG and BSA as MG covalently reacts with Arg and Lys side chains, and the scenario of being bounded to BSA through intermolecular interactions its highly unlikely to occur. Hence, the authors should remove the docking data involving AG and MG as it does not have any sense.

Response: Thank you very much for your observations. The present manuscript was carefully revised. We evaluated the interaction of MGO with BSA only for comparison with the remaining compounds that prevent glycation which is AG and ZnO-NPs. Effectively, our purpose was to visualize that the MGO, AG and ZnO-NPs interact with BSA in a similar binding site, as shown in Figure 8, focusing on AG and ZnO-NPs. Following the reviewer's suggestions, Fig. 8 was removed and replaced by a new one (Fig. 10).

Q18. In other hand, I think that the docking data arising from the use of ZnO-NPs is good and reports that a plausible mechanism of inhibition is the protection of Lys and Arg side chains. Nevertheless, the authors should clarify in the manuscript:

 (i) How they get the 3D structure of the ZnO-NPs that they used for the docking;

Response: We included the details of ZnO-NPs structure in the text:

Line 255-259, page 8: The characterization of the binding site between the BSA protein with AG and ZnO-NPs was estimated by Molecular Docking. For ZnO-NP docking, the structure reported by our previous studies was used [28]. Flat ZnO-NPs surfaces (15 Å x 25 Å) mimicking the particle curvature of a ZnO-NP were constructed. ZnO structure was generated using the GaussView software and Gaussian09 program package. The surfaces were optimized using MGL Tools 1.5.6 by adding Marsilli-Gasteiger partial charges on each constituent atom. The surface diameters were estimated using VEGA ZZ Support Pack. For AG molecule, their structure was previously optimized using Gaussian09 program package [29].

(ii) Which is the difference of between this and the structure of the chemically synthetized ZnO-NPs, which are not able to inhibit protein glycation. Would it be true that if the BSA-ZnO-NPs binding is the main mechanism of inhibition, the chemically synthetized ZnO-NPs should also be able to inhibit glycation?

Response: The differences between them are related to sizes and surfaces. Zinc oxide nanopowder has a size of nanoparticle <100 nm while the synthesized ZnO-NPs nanoparticles from M. indica have a size was ~6-12 nm. Nanoparticles can progressively and selectively adsorb biomolecules when they come into contact with complex biological fluids (M. indica in the present research). This surface, called ‘corona’ of biomolecules lowers the surface energy of the nanoparticle and promotes its dispersion and higher interaction with the biological system. According to the bibliography, these interactions lead to the formation of the protein corona, particle wrapping, intracellular uptake and biocatalytic process (NATURE NANOTECHNOLOGY, Vol 7, 2012, and DOI: 10.1038/NNANO.2012.207). On the other hand, computational simulations, though a very complicated task for the macromolecular assembly onto a ZnO-NPs surface, can complement the experimental research in an effective manner. Thus, a flat ZnO-NPs surface (15 Å x 25 Å) mimicking the particle curvature of ZnO-NPs was constructed, for that reason; we choose a nanoparticle surface to evaluate the interaction using molecular docking.

Ref: Monopoli, M. P., Åberg, C., Salvati, A., & Dawson, K. A. (2012). Biomolecular coronas provide the biological identity of nanosized materials. Nature nanotechnology, 7(12), 779.

Minor Concerns

Q1. Line 47. The authors assume that AGEs formation induce conformational changes on the protein structure. However, recent reports have proven that this does not necessarily occur (BMC Biochem. 2011 Aug 5;12:41; Biomacromolecules. 2014 Sep 8;15(9):3449-62). Hence, I request the authors to replace “conformational changes of the biomolecules” for “the loss of the biomolecular function”.

Response: As per the reviewer’s suggestion, the sentence has been modified accordingly in the revised manuscript.

Q2. Line 48. The authors should define in the abstract what is ZnO-NPs, as it appears there for the first time.

Response: As per the reviewer’s suggestion, the definition of ZnO-NPs has been added in the revised manuscript.

Q3. Lline 58 and 118. “Methyglyoxal” should not be in capitals nor “Bovine”. Please correct.

Response: As per the reviewer’s suggestion, the changes have made in the revised manuscript accordingly.

Q4. Line 59. The inhibition of ZnO-NPs on the glycation-induced conformational changes is not obvious at all, basically because it is not clear that glycation induced conformational changes. Please remove “glycation-induced conformational changes” and leave “glycation-induced aggregation”.

Response: As per the reviewer’s suggestion, the changes have made in the revised manuscript accordingly.

Q5. Line 62. Please define RBCs in the abstract.

Response: As per the reviewer’s suggestion, the definition of RBCs has been added in the revised manuscript.

Q6. Line 73. AGEs are not only formed under diabetic conditions but also linked to aging. Please, rephrase this sentence as it gives the idea that AGEs are only linked to diabetes mellitus.

Response: As per the reviewer’s suggestion, the sentence has been rephrased in the revised manuscript.

Q7. Line 75. Amino groups do not induce the sugar oxidation. Maillard reaction involves the reaction between reducing sugars and amino acid side chains. Hence, change “gets oxidized” by “reacts”.

Response: As per the reviewer’s suggestion, the changes have made in the revised manuscript accordingly.

Q8. Line 78. What the authors wanted to say with “the process of AGEs”? Would have been better to write “AGEs leads to….”. In addition, I suggest the authors to remove “structural” or change it by “chemical modifications”.

Response: As per the reviewer’s suggestion, the changes have made in the revised manuscript accordingly.

Q9. Line 80. The authors should include a reference proving that AGEs formation induce the formation of disulphur linkage. I never heard about it. If they are not able to provide such a reference, they must remove this sentence.

Response: As per the reviewer’s suggestion, a reference has been quoted in the revised manuscript wherein it has been stated that glycation induced modification have a detrimental impact on albumin structure as its conformation changes, free thiols gets oxidized to disulfide linkage or thiol radicals.

Ref: Bourdon, E., Loreau, N., & Blache, D. (1999). Glucose and free radicals impair the antioxidant properties of serum albumin. The FASEB Journal, 13(2), 233-244.

Q10. Line 81. Why b proteins? The authors must remove “beta” and leave proteins, in general.

Response: As per the reviewer’s suggestion, the changes have made in the revised manuscript accordingly.

Q11. Line 83.When the authors explain the AGEs inhibitors. Why they only focused in AG? There are other inhibitors with higher pharmacological potential, like pyridoxamine. The authors should slightly expand this and cover the most known AGEs inhbitors.

Response: As per the reviewer’s suggestion, the changes have made in the revised manuscript by the citation of references related to other AGEs inhibitors (Line 87-93).

Q12. Line 91. Replace “normal structure of macro molecules” by “native structure of biomolecules”.

Response: As per the reviewer’s suggestion, the changes have made in the revised manuscript accordingly.

Q13. Line 96-97. The authors claim that biosynthesized ZnO-NPs have better antimicrobial activity than chemically synthetized ZnO-NPs. However, the authors did not provide any reference proving it. Hence they should include one or remove the sentence.

Response: As per the reviewer’s suggestion, the sentence has been removed in the revised manuscript.

Q14. Line 126. The authors must better describe the experimental procedure that they followed to extract the components of the leafs. They should report the temperature, the time, if it was done under agitation, etc…

Response: As per the reviewer’s suggestion, the experimental procedure for extraction is given in the revised manuscript.

Q15. Line 140. The authors should remove “under in-vitro” since it is understood from the section itself, but also because in the first sentence they say that they used aminoguanidine “under in-vitro and in-vivo”. 

Response: As per the reviewer’s suggestion, the changes have made in the revised manuscript accordingly.

Q16. Line 144. This last sentence is part of the results, no part of the methodology. Please, move it to the result section.

Response: As per the reviewer’s suggestion, it has been moved to the results section in the revised manuscript.

Q17. Line 219. The authors state that blood glucose levels < 250mg/ml was taken as hyperglycaemic conditions. Is that correct? Should it be > 250mg/ml?

Response: Thank you very much for your keen observations. The error has been rectified in the revised manuscript.

Q18. Line 266. To be statistically significant, p should be < 0.05 no ≤ 0.05. Please correct.

Response: As per the reviewer’s suggestion, the changes have been made throughout the revised manuscript accordingly.

Q19. Line 272. Change “subjected to UV-Vis spectral analysis” by “analyzed by using UV-Vis spectroscopy”.

Response: As per the reviewer’s suggestion, the changes have made in the revised manuscript accordingly.

Q20. Line 277. Remove “as”. In addition, the authors should include the Scherrer’s formula, which would allow the reader to understand how they obtained the size.

Response: As per the reviewer’s suggestion, the Scherrer’s formula is incorporated in the Materials and Methods section (Line 154) in the revised manuscript accordingly.

Q21. Line 303. Would it be “the hemoglobin of RBCs” instead of “The RBCs of hemoglobin”?

Response: As per the reviewer’s suggestion, the changes have made in the revised manuscript accordingly.

Round 2

Reviewer 3 Report

The new version of the manuscript written by A et al., has notably improved from the original version. The changes that the authors have incorporated have clarified the main message and strengthened their findings. However, this reviewer still has some concerns that needs to be addressed before I can recommend the acceptance of the manuscript.

Q1. The authors have properly explained that they used δ-gluconolactone to specifically study the formation/effect of the Amadori products. In their rebuttal, they also state that “we also confirm that after 16 h incubation there will be no formation of AGEs yet”. However, this reviewer could not find any experimental evidence proving the lack of formation of AGEs. I find mandatory that the authors provide this information.

Q2. The authors did not properly respond to my original question (Q8). They did not explain in the rebuttal nor in the manuscript what is the specific process that they wanted to study by using MG. They neither explained the meaning of “intermediate state of glycation”, as I asked them. They should do it also in the manuscript.

On the other hand, they justify the use of MG because induced “oxidative stress”. The authors must explain which kind of oxidative stress can induce MG on a BSA solution!! MG is only a highly reactive carbonyl compound that rapidly reacts with protein side chains, but I doubt that can induce oxidative stress in vitro. From their reply I have the impression that they did not want to answer or they are not able to do it.

Q3.  The authors justify the use of the term “late state of glycation” in a scenario where the irreversible AGEs are formed. Do the authors think that MG does not induce the formation of AGEs? I is widely known that MG induces the formation of CEL and MOLD, in vivo but also in vivo. I think that this is not a proper way to explain it.

I have the impression that all these definition of early, intermediate and late state of glycation is quite confusing, and the entirely manuscript would be benefited from a complete redefinition. For instance, you could say that you use d-Glu to specifically study the effect of the formation of the Amadori compound. In addition, you could also justify the use of MG to specifically study the effect of a reactive dicarbonyl highly relevant in vivo, and the use of ribose as a highly reactive carbohydrate mimicking glucose, but also because increase its concentration in diabetes and it holds a notorious glycation potential per se (Wei et al., BBA-General Subjects (2012) 1820, 488-494).

Q4. The authors should know that the fact that somebody else wrongly used an enormously high concentration of ribose (0.8M) does not justify its use in further studies. I find it a really poor justification, especially if the authors do not bother themselves to properly check bibliography. The physiological concentration of ribose has been reported by Sun et al. (Sun T, et. al. Prog Biochem Biophys, 40, 816-825 (2013)), who proved that it notabily increases in diabetic mellitus. The authors should repeat some of the experiments to prove that they results are the same when using a 0.8M ribose concentration than when using the physiological one, or properly explain why they used 0.8M.

Q5. The authors have removed the word “quenching“ from the manuscript. However, this does not prevent that this could occurr during the experiments. I do not really understand why the authors do not performed the control experiment that I asked in the frist revision (titrate a solution containing Ac-Lys and Ac-Arg (in the same molar rations present in BSA) previously glycated with MG), which would only strength their fluorescent data. Instead, they just say that the fluorescent “data provided is from a reduction in the formation of AGEs and not the quenching effect”, without any experimental data proving it. They assessment must be supported by experimental data.

Q6. In their reply the authors explain that they studied the docking between MG and BSA just for comparison purposes. This should be clearly explained in the manuscript. Then they say that the “Figure 8 was removed and replaced by a new on (Fig. 10). However, I could not find this figure 10 in the manuscript. The authors must explain what they replaced and where is fig 10.
